# Quality Assessment of FY-3D/MERSI-II Thermal Infrared Brightness Temperature Data from the Arctic Region: Application to Ice Surface Temperature Inversion

**Haihua Chen \*, Xin Meng** 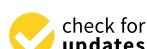**, Lele Li and Kun Ni**

Faculty of Information Science and Engineering, College of Marine Technology, Ocean University of China, Qingdao 266100, China
* Correspondence: chh7791@ouc.edu.cn; Tel.: +86-532-6678-2907

**Abstract:** The Arctic region plays an important role in the global climate system. To promote the application of Medium Resolution Spectral Imager-II (MERSI-II) data in the ice surface temperature (IST) inversion, we used the thermal infrared channels (channels 24 and 25) of the MERSI-II onboard Chinese FY-3D satellite and the thermal infrared channels (channels 31 and 32) of the Earth Observing System (EOS) Moderate-Resolution Imaging Spectroradiometer (MODIS) onboard the National Aeronautical and Space Administration (NASA) Aqua satellite for data analysis. Using the Observation–Observation cross-calibration algorithm to cross-calibrate the MERSI and MODIS thermal infrared brightness temperature ($T_b$) data in the Arctic, channel 24 and 25 data from the FY-3D/MERSI-II on Arctic ice were evaluated. The thermal infrared $T_b$ data of the MERSI-II were used to retrieve the IST via the split-window algorithm. In this study, the correlation coefficients of the thermal infrared channel $T_b$ data between the MERSI and MODIS were >0.95, the mean bias was −0.5501–0.1262 K, and the standard deviation (Std) was <1.3582 K. After linear fitting, the MERSI-II thermal infrared $T_b$ data were closer to the MODIS data, and the bias range of the 11 μm and 12 μm channels was −0.0214–0.0119 K and the Std was <1.2987 K. These results indicate that the quality of the MERSI-II data is comparable to that of the MODIS data, so that can be used for application to IST inversion. When using the MERSI thermal infrared $T_b$ data after calibration to retrieve the IST, the results of the MERSI and MODIS IST were more consistent. By comparing the IST retrieved from the MERSI thermal infrared calibrated $T_b$ data with MODIS MYD29 product, the mean bias was −0.0612–0.0423 °C and the Std was <1.3988 °C. Using the MERSI thermal infrared $T_b$ data after calibration is better than that before calibration for retrieving the IST. When comparing the Arctic ocean sea and ice surface temperature reprocessed data (L4 SST/IST) with the IST data retrieved from MERSI, the bias was 0.9891–2.7510 °C, and the Std was <3.5774 °C.

**Keywords:** MERSI; MODIS; brightness temperature; quality assessment; ice surface temperature

## 1. Introduction

The stable Arctic ice sheet covers more than one-third of the surface of the Arctic Ocean, in addition to Greenland, other Arctic islands and high-latitude continental areas. Arctic sea ice plays an important role in the global climate system and global climate change. Sea ice forms in the Arctic when the temperature of seawater is below −1.8 °C. Sea ice affects tides and tidal currents, reducing both the tidal range and tidal velocity. Sea ice also reduces the wave height and impedes wave propagation. The ice surface temperature (IST) controls the speed of growth, the melting of ice, and the exchange of heat between the ocean and the atmosphere [1]. The IST can be used to observe changes in the extent and thickness of sea ice in the Arctic and is an important parameter in sea ice models.

IST has been derived from infrared radiation (IR) channel data on various satellites [2], such as the Advanced Very High Resolution Radiometer (AVHRR), the Visible Infrared

Imaging Radiometer Suite (VIIRS) and the Earth Observing System (EOS) Moderate-Resolution Imaging Spectroradiometer (MODIS). We used the MODIS L2 MYD29 IST product [3], which was obtained by the split-window algorithm [4] with the 11 and 12 μm thermal infrared brightness temperature ($T_b$) data. The IST products based on MODIS are widely used, and the algorithms have the same form and accuracy as the AVHRR IST products [5].

The FengYun-3 (FY-3) meteorological satellite is the second generation of polar-orbiting meteorological satellites launched by China, and it carries the Medium Resolution Spectral Imager (MERSI). The spatial resolution and band settings of FY-3D/MERSI are basically consistent with those of MODIS onboard the National Aeronautical and Space Administration (NASA) Aqua satellite (Aqua/MODIS). The FY-3D/MERSI thermal infrared channels 24 (10.26–11.26 μm) and 25 (11.50–12.50 μm) correspond to Aqua/MODIS channels 31 (10.780–11.280 μm) and 32 (11.770–12.270 μm). To date, there are no published IST products from FY satellites, so we evaluated the data quality of channels 24 and 25 of FY-3D/MERSI, and compared channel 11 μm and 12 μm data from Aqua/MODIS and FY-3D/MERSI to determine whether FY-3D/MERSI can be used to retrieve the IST. The deviation was corrected by cross-calibration between channels to ensure the reliability, accuracy, and consistency of the IST inversion data [6].

To promote the application of MERSI-II data in IST inversion, it is necessary to evaluate the data quality in the Arctic sea ice area. In 2004, Wan et al. [7] evaluated the MODIS thermal infrared bands and the status of land surface temperature (LST) version-3 standard products retrieved from Terra MODIS data. Liu et al. [8] compared VIIRS v1 LST with ground in situ observations and heritage LST products from MODIS Aqua and the Advanced Along-Track Scanning Radiometer (AATSR) in 2015. The cross comparison indicates an overall close LST estimation between VIIRS and MODIS. In order to promote the application of MERSI-II data in the ocean, Zhang and Qiu [9] used the VIIRS as a reference and preliminarily evaluated the quality of MERSI-II data from the aspects of signal-to-noise ratio (SNR). The results show that the MERSI-II data quality is comparable to that of VIIRS, so the data can be used in ocean remote sensing applications. Accurate radiometric calibration is an important measure and a guarantee for sensor data quality. Cross-calibration is a radiometric calibration method in which the orbiting satellite sensor to be calibrated observes the same target at the same time as the orbiting satellite sensor with previously established good calibration results. In 2016, Li et al. [10] cross-calibrated FY-3A/Visible and InfraRed Radiometer (VIRR) channel 4 with the high-accuracy MODIS channel 31 onboard NASA's Terra satellite and verified that the two channels had good consistency. Wang et al. [11] (2017) applied a snow depth service algorithm to cross-calibrate the Micro-Wave Radiation Imager (MWRI) onboard the FY-3B and FY-3D satellites to ensure the reliability and consistency of the MWRI brightness temperature data and to integrate binary MWRI observation data. In 2020, Tang and Chen [12] proposed the monthly cross-calibration of the brightness temperature data from the FY-3B/MWRI and the Advanced Microwave Scanning Radiometer 2 (AMSR-2) on the Global Change Observation Mission 1st-Water (GCOM-W1) in the Arctic region. In 2021, Chen et al. [13] proposed the monthly cross-calibration of the brightness temperature data of the FY-3B/MWRI and the Advanced Microwave Scanning Radiometer for Earth Observing System (AMSR-E) sensors in the Arctic region and compared the results of the observed snow depth with the results from the calibrated MWRI and AMSR-E products.

Thermal infrared channels data are widely used to obtain temperature data on land or sea surfaces, but it is difficult to obtain the IST, especially in Polar Regions where it is difficult to measure the actual temperature [14]. The split-window algorithm is the most mature algorithm used for surface temperature inversion based on two adjacent thermal infrared channels. Key J.R. and Haefliger [4] developed an IST retrieval algorithm for the Arctic in 1992 using thermal infrared data from channels 4 and 5 of the AVHRR onboard the National Oceanic and Atmospheric Administration (NOAA) polar-orbiting satellites. They validated the IST algorithm in 1994 using a set of field data collected during

the May–June 1992 Seasonal Monitoring and Simulation of Sea Ice field activities, which had previously been published for satellite IST inversion [15]. In 1997, they showed that the algorithm was accurate enough for most studies of climate processes. However, it was only applicable in clear sky conditions, and the influence of clouds could lead to significant errors in the calculation of the IST. They therefore modified the coefficients of the algorithm to enhance its practicality. This method also has reference values for the estimation of coastal land temperatures. These coefficients have been applied to the MODIS IST algorithm, and the accuracy of the IST is within the range of 0.3–2.1 K [16]. In 2014, Jiménez-Muñoz J.C. et al. [17] proposed the universal split-window algorithm, in which the atmospheric water vapor content was directly involved in the calculation. In 2015, Jin M.J. et al. [18] proposed a split-window algorithm that directly approximated the radiation transfer equation. Du C. et al. [19] adopted the equations used by the MODIS land temperature product, and used the practical split-window algorithm to estimate the land surface temperature from the Landsat 8 thermal infrared sensor and grouped the coefficients according to the water vapor content. The reliability of the algorithm has been verified by applications in different regions.

The FY-3D and Aqua are both afternoon satellites, and therefore their sensors can be used in comparisons. The MODIS ISTs have been compared with the near-surface air temperature in the Arctic Ocean obtained from the NOAA National Ocean Service (NOS) Center for Operational Oceanographic Products and Services (CO–OPS) Alaska tide stations and from drifting buoys from the North Pole Environmental Observatory buoy program [5]. The comparison results show that the mean bias was −2.1 °C and the root mean square error (RMSE) was 3.7 °C [5]. Through preprocessing, the MERSI and MODIS thermal infrared $T_b$ data at 11 µm and 12 µm were obtained. The MERSI thermal infrared channel $T_b$ data were compared and analyzed to those of the MODIS in the Arctic region (60°N–90°N, 180°W–180°E). The thermal infrared channel $T_b$ data at the 11 µm and 12 µm channels of MERSI-II are in good agreement with MODIS data after calibration. The two thermal infrared channel $T_b$ data of MERSI are further used to retrieve the IST using the split-window algorithm in the Arctic, and compared with the MODIS MYD29 products. We further validated the MERSI IST data via the Arctic Ocean sea and ice surface temperature reprocessed data (L4 SST/IST).

## 2. Materials and Methods

### 2.1. Datasets

The datasets contain the thermal infrared channel $T_b$ data observed by FY-3D/MERSI and Aqua/MODIS from November 2020 to December 2021 in the Arctic region (north of 60 °N). Because the data in November and December cannot be used due to some abnormal days, the data in November and December are the comprehensive data of 2020 and 2021. In this study, we also used IST data from the MYD29 datasets, L4 IST data, and sea ice concentration (SIC) data retrieved by FY-3D/MWRI brightness temperature.

#### 2.1.1. Aqua/MODIS Data

The afternoon-orbiting Aqua satellite was launched on 4 May 2002 as part of the NASA-centered international EOS [20]. Aqua's Sun-synchronous, near-polar orbit crosses the equator from south to north at about 1:30 pm local time. Global coverage occurs every one to two days (more often near the poles). MODIS is one of the sensors carried on the Aqua satellite. It has 36 channels with three different spatial resolutions of 250, 500, and 1000 m, including 20 visible to shortwave infrared channels and 16 thermal infrared channels. The scanning width is 2330 km with high sensitivity and accuracy [21]. We used the L1B calibrated radiance data of channels 31 (10.780–11.280 µm) and 32 (11.770–12.270 µm) of the Aqua/MODIS system provided in the MYD021 datasets, the IST product data from the MYD29 datasets, and the latitude and longitude data from the MYD03 datasets (Table 1).

The MODIS L2 product MYD29 is generated using the sensor radiation data products (MYD021), the geolocation products (MYD03), and the cloud mask products (MYD35_L2) [22]. The five-minute data include the sea ice extent, the IST data, and the mass estimates of the sea ice extent and the IST at a resolution of 1 km. This product has been verified in the Arctic Ocean with a high accuracy and can be used as comparative data. The MYD29 data are provided in HDF-EOS2 format and stored as an 8-bit unsigned integer.

### 2.1.2. FY-3D/MERSI Data

FY-3D, the fourth satellite in Chinese new generation of polar-orbiting FY-3 satellites, was successfully launched in November 2017. The satellite carries the second-generation MERSI-II, which includes nineteen 1 km resolution channels and six 250 m resolution channels. The MERSI-II thermal infrared channels 24 (10.26–11.26 μm) and 25 (11.5–12.5 μm) are split from a 250 m resolution spectral channel of MERSI-I with a central wavelength of 11.25 μm, and its ground resolution is unchanged at 250 m. The two thermal infrared channels (Table 1) [23] can be used for IST inversion using the split-window algorithm. The channel imaging capability is superior to that of MERSI-I and is at the international advanced level [24]. We evaluated MERSI-II data in the Arctic region and cross-calibrated them with MODIS thermal infrared channel $T_b$ data. Additionally, we used the MERSI thermal infrared channel $T_b$ data for IST inversion.

**Table 1.** Sources and resolutions of the data [21,23,25].

| Source | Datasets | Parameters | Central wavelength (μm) | Spectral bandwidth (nm) | Spatial Resolution (m) |
|---|---|---|---|---|---|
| NASA LAADS DAAC | MODIS | Channel 31 brightness temperature | 11.03 | 500 | 1000 |
| | | Channel 32 brightness temperature | 12.02 | 500 | 1000 |
| | MYD29 | Sea ice surface temperature | | | 1000 |
| | MYD03 | Latitude and longitude | | | 1000 |
| China National Satellite Meteorological Center | MERSI | Channel 24 brightness temperature | 10.8 | 1000 | 250 |
| | 1KM | Channel 25 brightness temperature | 12.0 | 1000 | 250 |
| | 1KM_GEO | Latitude and longitude | | | 1000 |

### 2.1.3. FY-3D/MWRI Arctic SIC Data

The SIC data were retrieved by domestic satellite FY-3D/MWRI brightness temperature, which was downloaded from the Key Lab of Polar Oceanography and Global Ocean Change (http://coas.ouc.edu.cn/pogoc/sy/list.htm). The SIC inversion algorithm is a Dynamic Tie point ASI algorithm (DT-ASI algorithm) [26] that was developed on the basis of the ASI algorithm of Bremen University (UB) in Germany. The brightness temperature data has been cross-corrected [27,28]. The spatial resolution of the data is 12.5 km and the time resolution is the daily average. The IST algorithm is constrained to ocean pixels that are not obstructed by clouds and is run for daytime and nighttime data [21]. When the SIC is less than 15%, it is regarded as water. Thermal infrared channels $T_b$ and IST data

from MERSI and MODIS were obtained by extracting sea ice areas with SIC data greater than 15%.

### 2.1.4. Arctic Ocean Sea and Ice Surface Temperature Reprocessed Data

The validation data in this paper were the Arctic Ocean sea and ice surface temperature reprocessed data (L4 SST/IST) from 1982 to May 2021 (>58 °N), with the spatial resolution 0.05°. The L4 IST data were compared with the Heitronics KT19.85 Series II Infrared Radiation Pyrometer (KT−19) measurements from Icebridge Flight, showing a mean difference of 1.52 °C, and with air temperatures from Arctic ice drifting stations, European Centre for Medium-Range Weather Forecasts (ECMWF) distributed buoys and Cold Regions Research and Engineering Laboratory (CRREL) buoys, and the mean differences were −2.35 °C, −3.21 °C, and −2.87 °C, respectively [29]. The L4 SST/IST data were released by the Copernicus Marine Service (https://data.marine.copernicus.eu/product/SEAICE_ARC_PHY_CLIMATE_L4_MY_011_016/description (accessed on 16 November 2022)). In this paper, the L4 IST data were compared with the IST data retrieved from the MERSI thermal infrared channel $T_b$ data after calibration.

### 2.2. Methods

### 2.2.1. Data Preprocessing

The central wavelengths of MERSI-II thermal infrared channel 24 and MODIS thermal infrared channel 31 are 10.8 μm and 11.03 μm, respectively. The central wavelengths of MERSI-II thermal infrared channel 25 and MODIS thermal infrared channel 32 are 12.0 μm and 12.02 μm, respectively (Table 1). In the spectral response function (SRF) (Figure 1), we compared the 11 μm and 12 μm channels, and obtained the cross of the MODIS spectral range and the MERSI spectral range. However, due to the different spectral bandwidths, the band range is still different, so it is necessary to compare the 11 μm and 12 μm channels data of MODIS and MERSI.

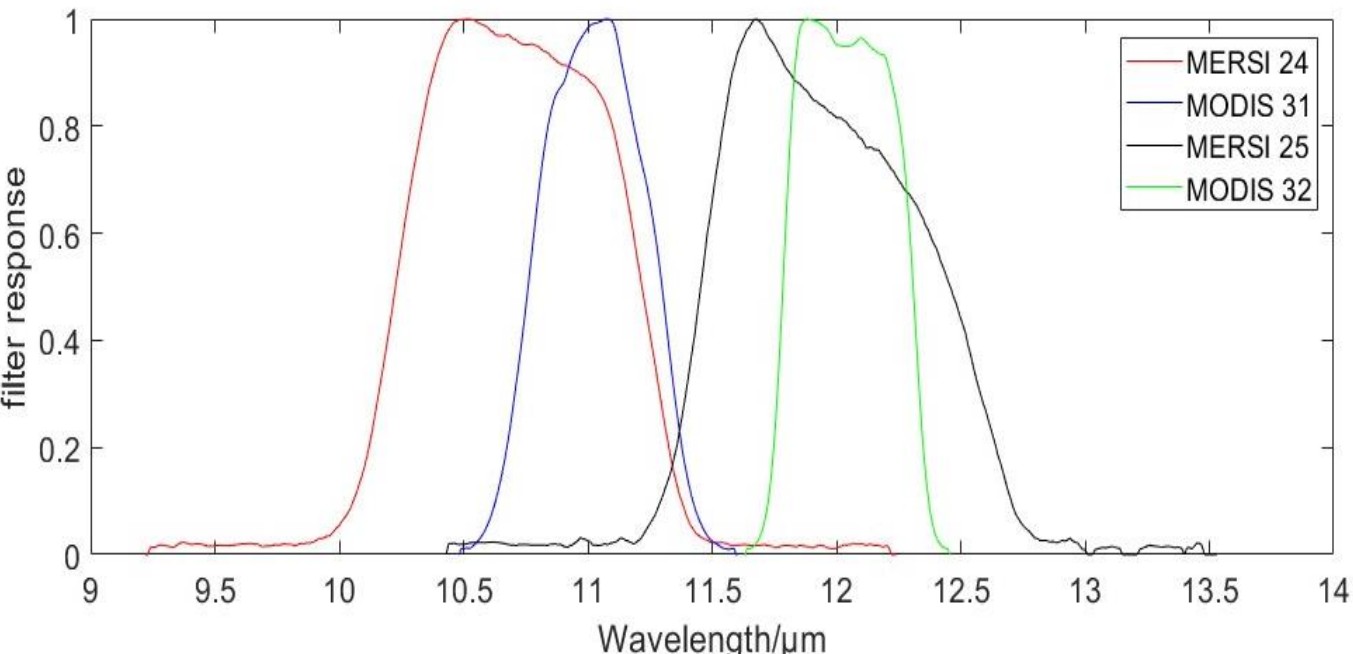

**Figure 1.** MERSI (channels 24 and 25) and MODIS (channels 31 and 32) central wavelengths at 11 μm and 12 μm spectral response function.

Aqua/MODIS Thermal Infrared Channels Brightness Temperature

The MYD021 datasets include the 250 m and 500 m resolution bands aggregated to a 1 km resolution. The radiation $L_i$ (mW/(m$^2$ cm sr)) of thermal infrared channels 31 and 32 is calculated as follows [25]:

$$L_i = radiance\_scales_i \times (DN_i - radiance\_offset_i) \tag{1}$$

where $DN_i$ is the raw digital signals measured at the detectors; $radiance\_scales_{31}$ = 6.508072 × 0.0001, $radiance\_scales_{32}$ = 5.7100126 × 0.0001, $radiance\_offset_{31}$ = 2035.9332, and $radiance\_offset_{32}$ = 2119.0845 are used to the calculate radiance [25].

Through Planck's inverse transformation formula, the radiance of the black body $L(T,\lambda)$ (mW/(m$^2$ cm sr)) obtained in Equation (1) is used to calculate the brightness temperature ($T_{b31}$, $T_{b32}$) of channels 31 and 32.

The polar projection is performed on the obtained $T_b$ data of the MODIS channels 31 and 32, the latitude and longitude data and the IST data to produce a grid with a resolution of 4 km.

FY-3D/MERSI Thermal Infrared Channels Brightness Temperature

The amplification radiance value $RAD0$ in the MERSI data was calibrated and calculated using Equation (2), where slope and intercept are attributes in the MERSI datasets. The radiance $RAD$ (mW/(m$^2$ cm sr)) of channels 24 and 25 was obtained as follows [23]:

$$RAD = RAD0 \times slope + intercept \tag{2}$$

where $slope$ = 0.01 and $intercept$ = 0.0.

We use the Planck's inverse transformation formula to convert radiance $RAD$ to the equivalent black-body brightness temperature $Te_i$ (K). Channel brightness temperature correction coefficients $A_i$ and $B_i$ are used to convert $Te_i$ into channel black-body brightness temperature $T_{b24}$ and $T_{b25}$ as follows [23]:

$$T_{b24} = A_{24} \times Te_{24} + B_{24} \tag{3}$$

$$T_{b25} = A_{25} \times Te_{25} + B_{25} \tag{4}$$

where the brightness temperature correction coefficients of channel 24 are $A_{24}$ = 1.00133, $B_{24}$ = −0.0734 and for channel 25 are $A_{25}$ = 1.00065, $B_{25}$ = 0.0875.

### 2.2.2. Cross-Calibration Method

There are three main methods of cross-calibration [30]: (1) Radiative Transfer Model (RTM)-based calibration (Calibration–Observation, C–O), which is suitable when there is no overlapping observation band; (2) observation-to-observation correction (Observation–Observation, O–O), where the two sensors are required to have the same physical configuration (e.g., frequency or resolution) and the same observation overlap band; and (3) the double-difference mode correction (Difference–Difference, D–D), where the model simulation or a third sensor is taken as the reference target to calculate and compare the differences from the reference target. These methods can all reduce the influence of different sensor configurations.

Both FY-3D and Aqua are afternoon satellites with a temporal resolution of five minutes. The spatial resolution of MODIS channels 31 and 32 are 1 km, and that of MERSI channels 24 and 25 are 250 m, which were fused to 1 km. In the Arctic region, the two satellites have overlapping regional observation data within five minutes, so cross-calibration can be carried out based on O–O correction. The thermal infrared $T_b$ data of the MERSI and MODIS have a similar overall distribution and strong linear correlation. The linear equations were used for cross-calibration based on the least squares method. We find the best function match of the data by minimizing the sum of the squares of the errors between the MODIS and MERSI thermal infrared channel $T_b$ data. Using the

brightness temperature of MODIS channels 31 and 32 as the reference, channels 24 and 25 of MERSI are corrected, and the calibrated $T'_{b24}$ and $T'_{b25}$ are obtained, as shown in Equations (5) and (6):

$$T'_{b24} = K_1 T_{b24} + b_1 \tag{5}$$

$$T'_{b25} = K_2 T_{b25} + b_2 \tag{6}$$

where $K_i$ is the slope and $b_i$ is the intercept in the equation of cross-calibration between the MERSI and MODIS data.

### 2.2.3. Split-Window Algorithm

IST is estimated using a split-window algorithm and a version of Key's equation for the AVHRR that was adapted for use with MODIS channels 31 and 32 [16,21]:

$$\text{IST} = a + bT_i + c(T_i - T_j) + d\big((T_i - T_j)(\sec(q) - 1)\big) \tag{7}$$

where $T_i$ is the thermal infrared channel $T_b$ data at 11 μm, corresponding to the two thermal infrared channel $T_b$ data of channel 24 (10.26–11.26 μm) of MERSI and channel 31 (10.780–11.280 μm) of MODIS; $T_j$ is the thermal infrared channel $T_b$ at 12 μm, corresponding to the two thermal infrared channels' $T_b$ data, MERSI channel 25 (11.50–12.50 μm) and MODIS channel 32 (11.770–12.270 μm); $q$ is the satellite zenith angle, and $a$, $b$, $c$ and $d$ in Table 2 are the coefficients between the brightness temperature and the estimated IST determined by multiple linear regression [21].

**Table 2.** Coefficients determined by multiple linear regression between MODIS brightness temperature and the estimated surface temperature in the northern hemisphere [21].

| Northern Hemisphere | *a* | *b* | *c* | *d* |
|---|---|---|---|---|
| $T < 240$ K | 1.5711228087 | 1.0054774067 | 1.8532794923 | 0.7905176303 |
| 240 K $< T <$ 260 K | 2.03726968515 | 1.0086040702 | 1.6948238801 | 0.2052523236 |
| $T > 260$ K | 4.2953046345 | 1.0150179031 | 1.9495254583 | 0.197132579 |

The algorithm is applied to all ocean pixels without pre-screening for the possible occurrence of sea ice [21]. Water vapor and the presence of any clouds have the potential to reduce the accuracy of the IST [22]. Under ideal conditions (clear skies and low water vapor), the IST accuracy provided by the MYD29 product is estimated to be 1–3 °C [2,5,21,31].

### 2.2.4. Accuracy Evaluation Index

In this research, several indicators were used for statistical analysis, including the mean bias, the standard deviation (Std) and the correlation coefficient (Corr) of the brightness temperature between the MERSI and MODIS before and after calibration. The formulas are defined as follows:

$$E_{\text{std}} = \sqrt{\frac{1}{N-1} \sum_{i=1}^{N} \left( X_i - \overline{X} \right)^2} \tag{8}$$

$$E_{\text{bias}} = \frac{\sum_{i=1}^{N} (X_i - Y_i)}{N} \tag{9}$$

where $N$ represents the number of samples, the subscript $i$ represents a data point, and $X$ and $Y$ are the two estimated quantities.

## 3. Results

### 3.1. Brightness Temperature Preprocessing Results

The five-minute thermal infrared channel $T_b$ data from MERSI and MODIS were obtained using the preprocessing method in Section 2.2.1. The channel 24 and 25 $T_b$ data, longitude and latitude data, and zenith angle data from MERSI and channel 31 and 32 $T_b$ data, longitude and latitude data, and IST data from MODIS were projected to the $1647 \times 1647$ grid with a resolution of 4 km, respectively.

The period of MERSI and MODIS strip data is 5 minutes, and the spatiotemporal matching of data between the two data of the Arctic region (60°N–90°N, 180°W–180°E) was carried out in this study. Taking 19:50 on 2 January 2021 as an example, Figure 2 shows the strip data from MERSI and MODIS, and Figure 3 shows that MERSI and MODIS data display overlapping geographical locations in the same period after spatial matching.

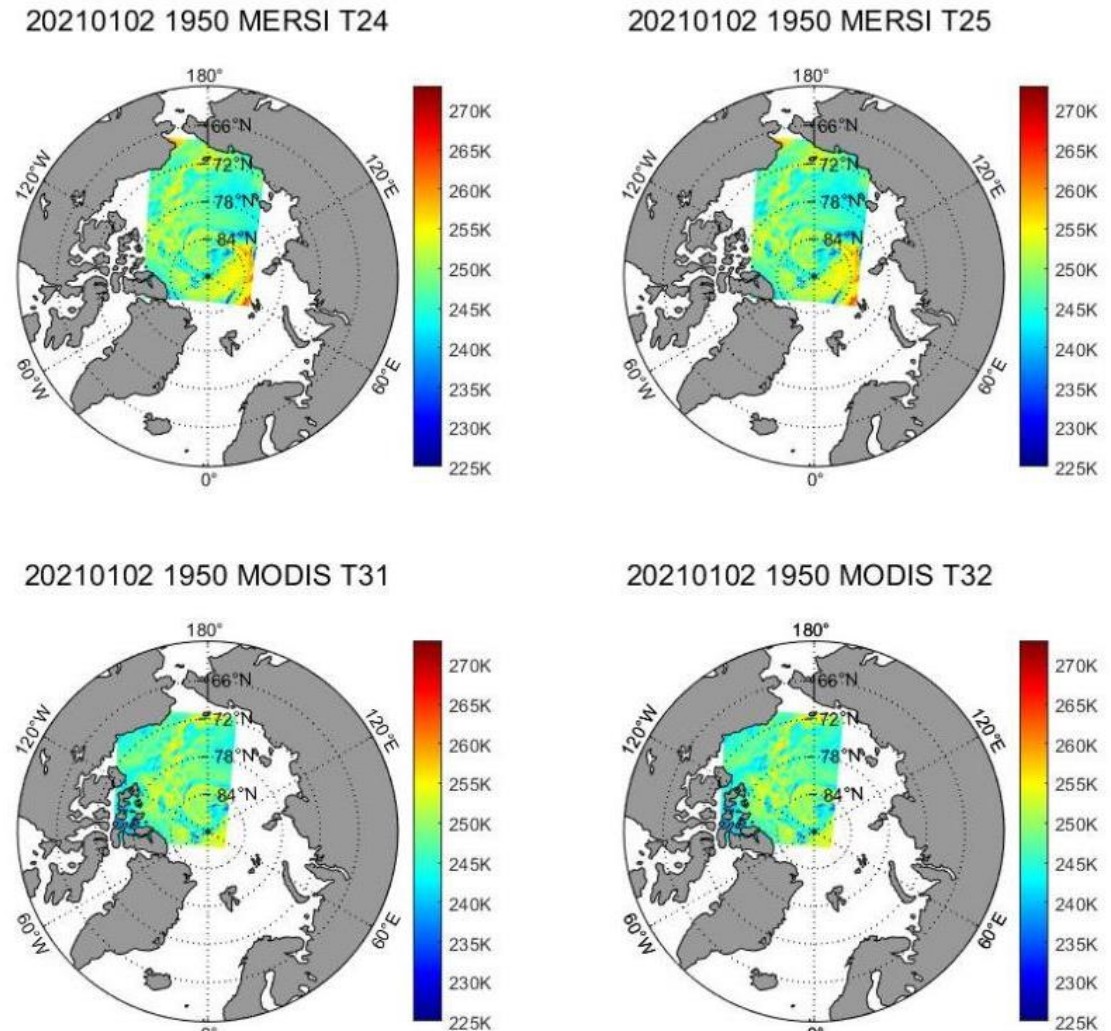

**Figure 2.** MERSI (channels 24 and 25) and MODIS (channels 31 and 32) thermal infrared $T_b$ data before spatial matching at 19:50 on 2 January 2021.

We controlled the quality of the data by removing the water with the SIC data and limiting the IST to below −1.8 °C while removing clouds with the MODIS cloud mask. In this paper, clouds are masked using the MODIS Cloud Mask (MYD35_L2) 'unobstructed field-of-view' flag. The flag includes 'cloudy', 'uncertain clear', 'probably clear', and 'confident clear'. If the flag is set to 'certain cloud', the pixel is set to 'cloud'. If the cloud flag is set to 'clear', or any level of possible cloud, the pixel is interpreted as 'clear' [21]. In

order to increase the number of retrievals balanced against the cloud conservative nature of the cloud mask, we accept the potential for ice/cloud confusion and cloud contamination in IST.

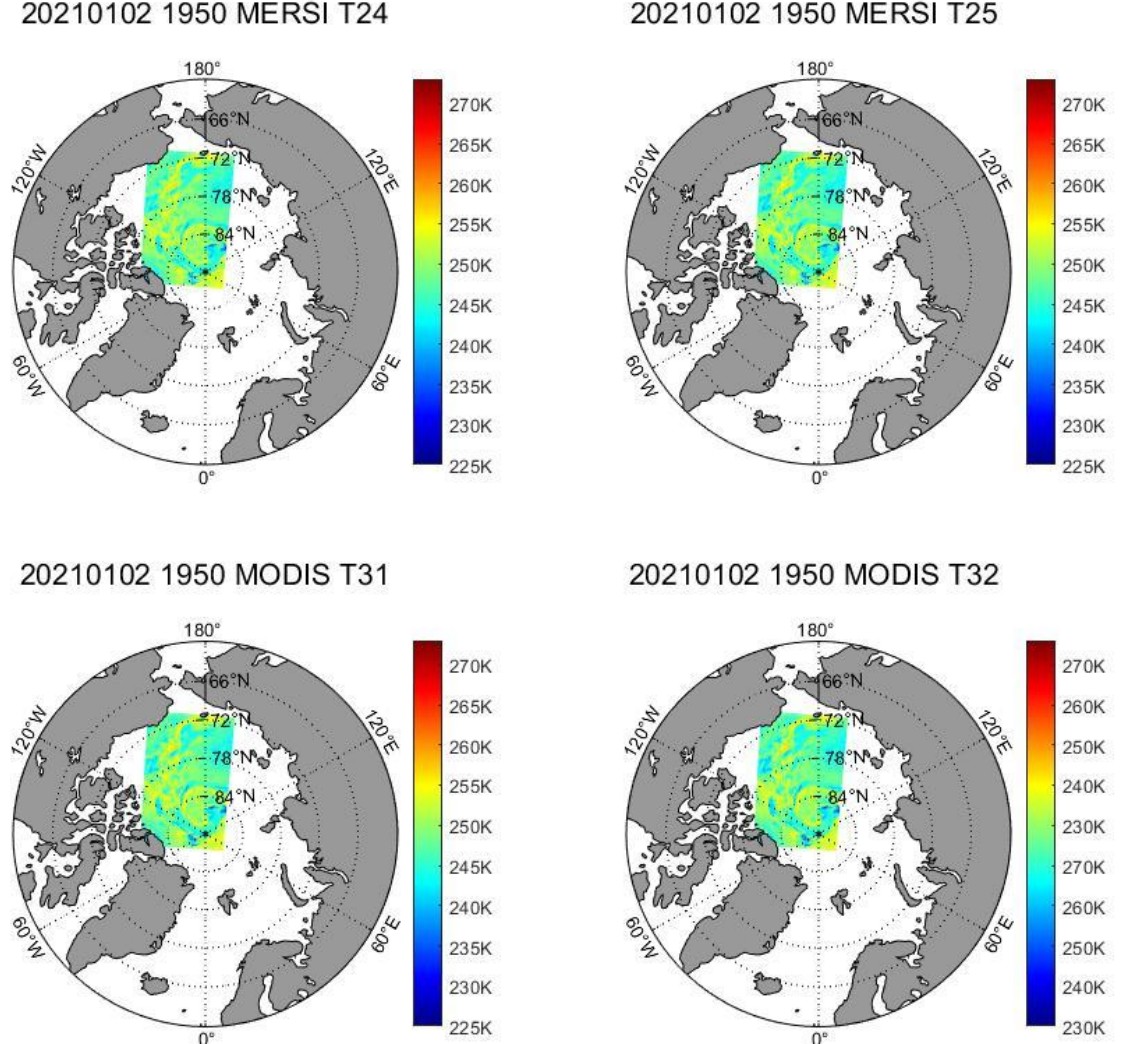

**Figure 3.** MERSI (channels 24 and 25) and MODIS (channels 31 and 32) thermal infrared $T_b$ data after spatial matching at 19:50 on 2 January 2021.

In the Arctic, the $T_b$ data were averaged to obtain the daily average thermal infrared 11 μm and 12 μm channel $T_b$ data for MERSI and MODIS (Figure 4). The white areas without data in Figure 4 are water, clouds, and some outliers, and the remaining areas with data are ice. According to the distribution map, there are some differences between MERSI and MODIS thermal infrared channel $T_b$ data, but the overall distribution was mostly consistent.

### 3.2. FY-3D/MERSI and Aqua/MODIS Cross-Calibration Results

In this research, we found that there were abnormal data in the MERSI data, so the abnormal data from FY-3D/MERSI were removed during the subsequent calculation process. We used the O–O method to cross-calibrate the MERSI and MODIS thermal infrared channel $T_b$ data and obtained the cross-calibration linear equation. Figure 5 shows the scatter plots diagram of MERSI and MODIS thermal infrared channel $T_b$ data from January to December. The regression equations (the top of each sub-graph) show a clear linear relationship between the MERSI and MODIS data.

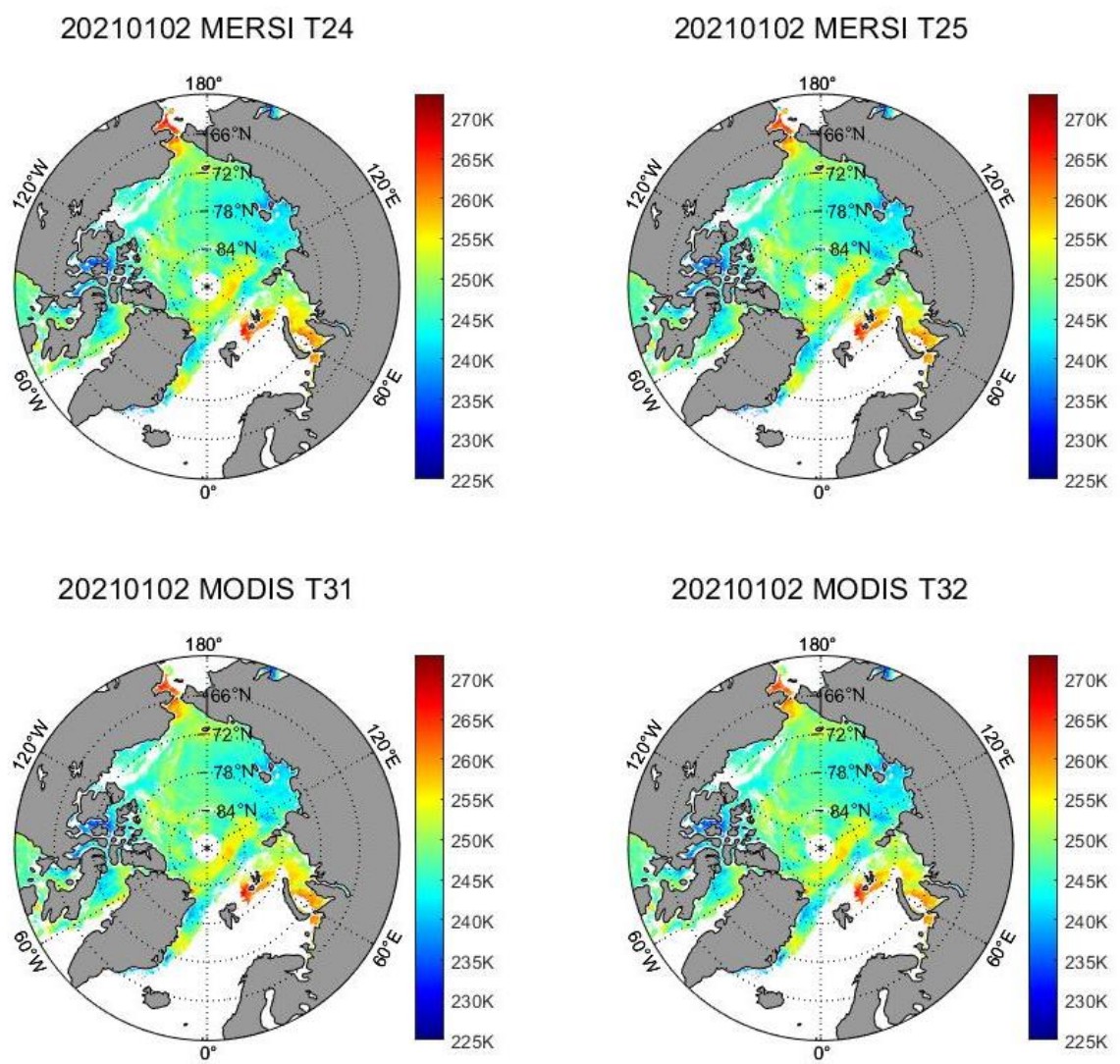

**Figure 4.** Comparison of distribution maps of MERSI (channels 24 and 25) and MODIS (channels 31 and 32) daily average thermal infrared $T_b$ data on 2 January 2021.

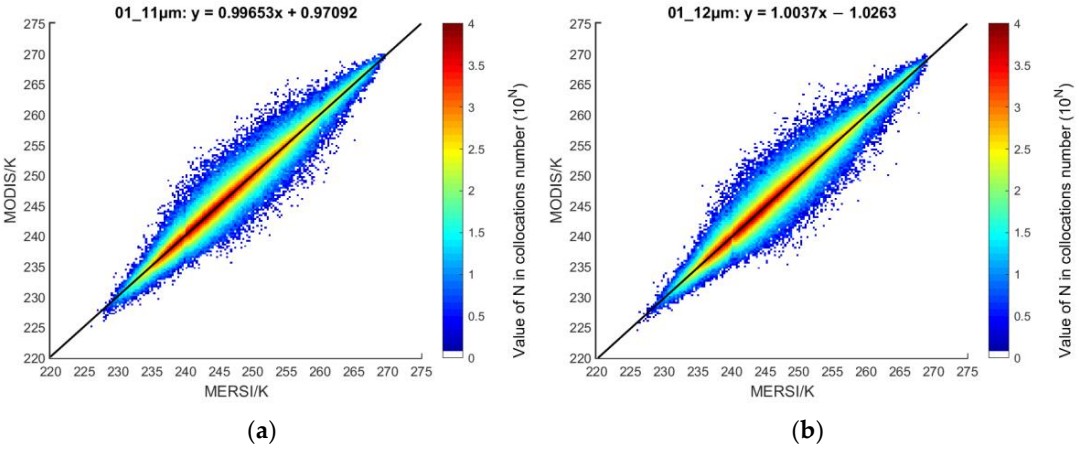

| (**a**) | (**b**) |

**Figure 5.** *Cont.*

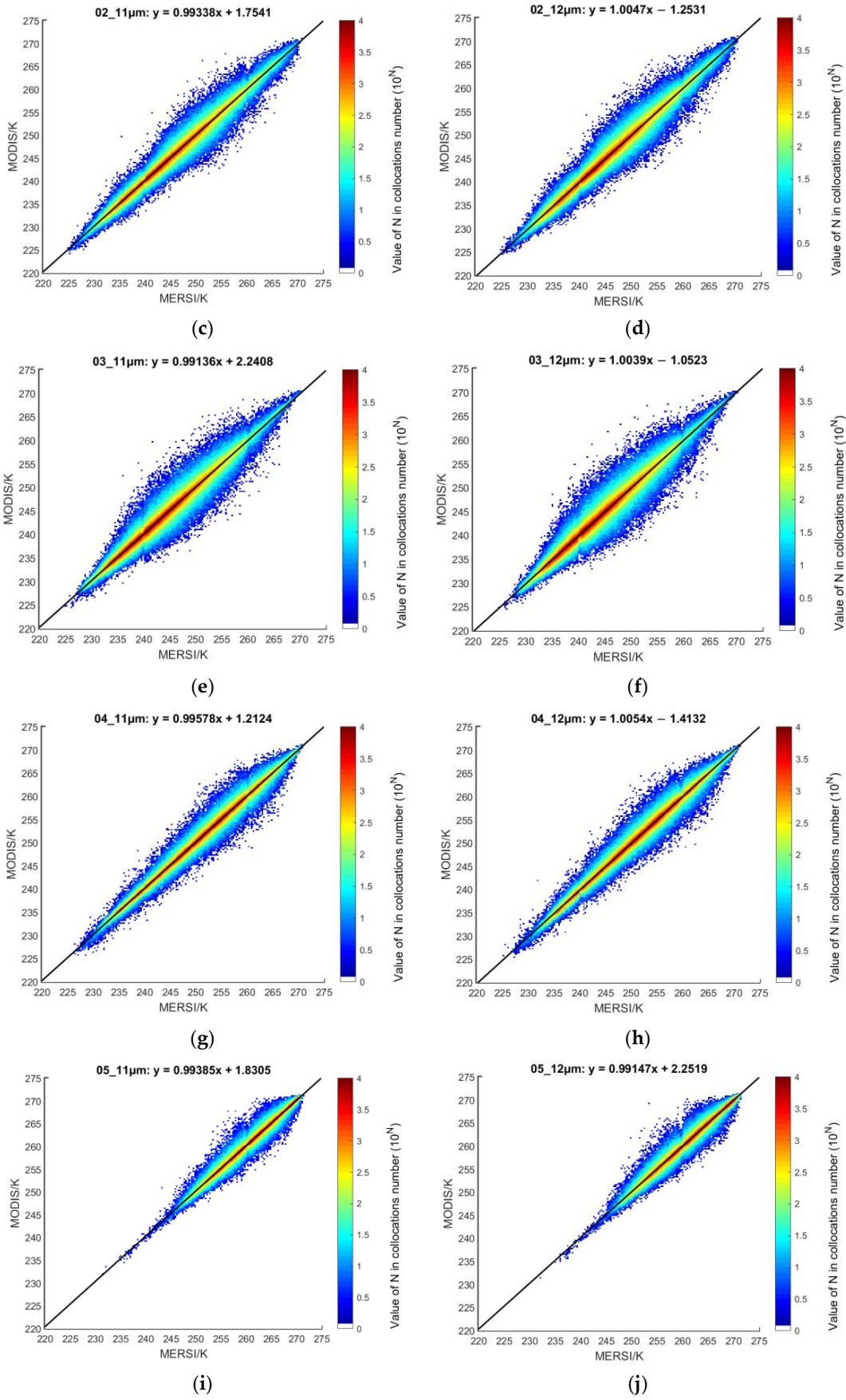

**Figure 5.** *Cont.*

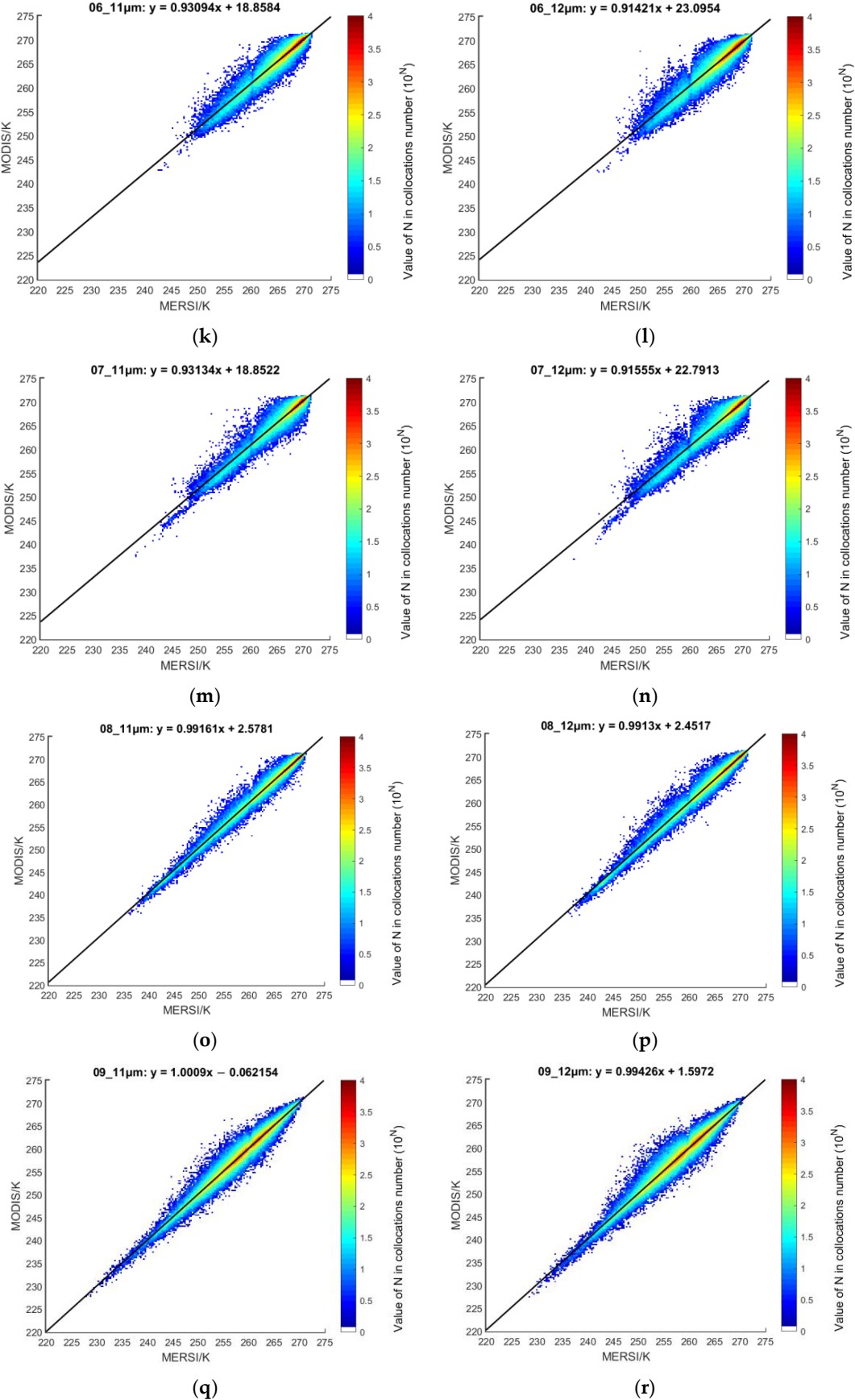

**Figure 5.** *Cont.*

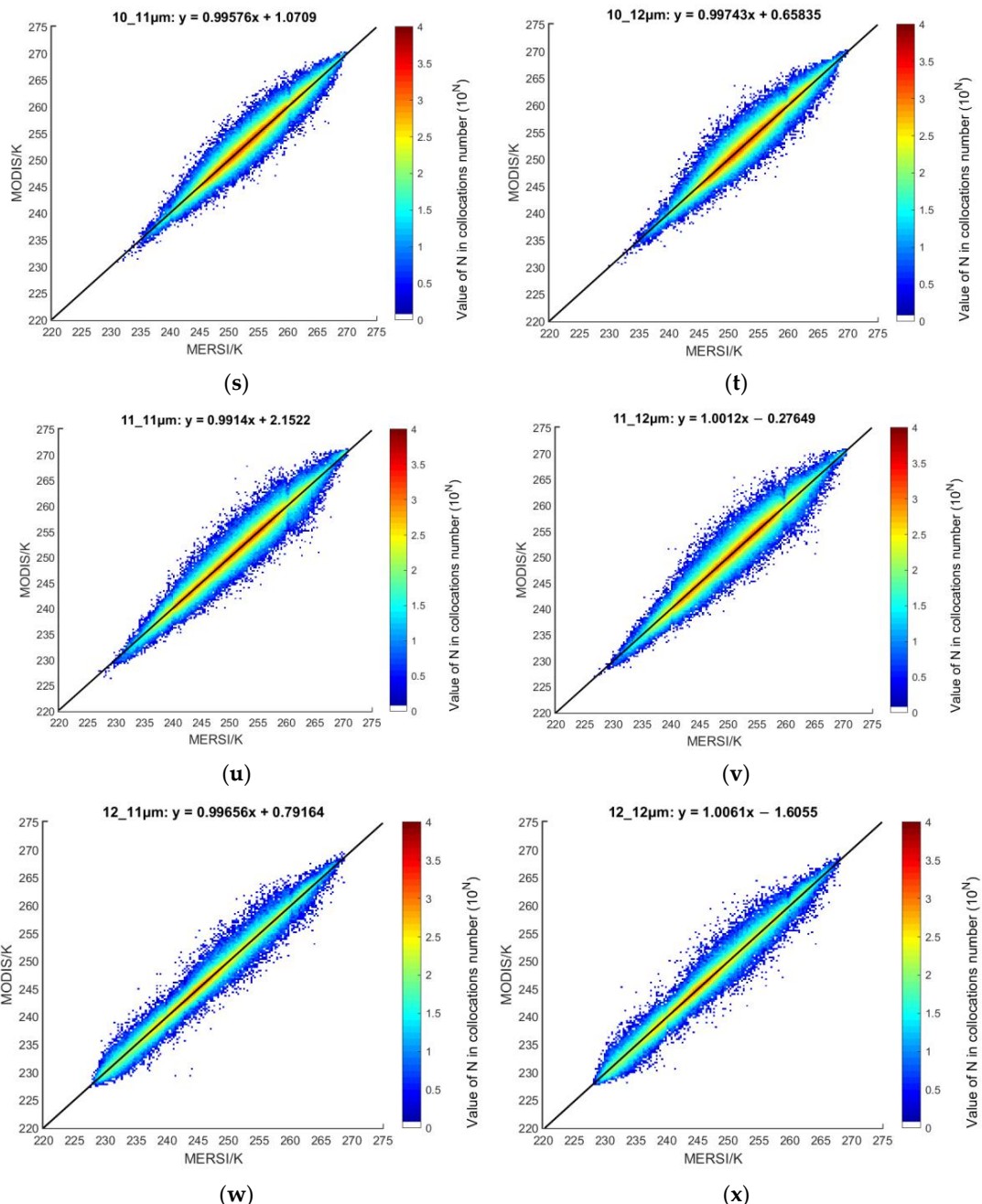

**Figure 5.** Cross-calibration of the FY-3D/MERSI and Aqua/MODIS thermal infrared $T_b$ data for each month. (**a**) 11 January μm. (**b**) 12 January μm. (**c**) 11 February μm. (**d**) 12 February μm. (**e**) 11 March μm. (**f**) 12 March μm. (**g**) 11 April μm. (**h**) 12 April μm. (**i**) 11 May μm. (**j**) 12 May μm. (**k**) 11 June μm. (**l**) 12 June μm. (**m**) 11 July μm. (**n**) 12 July μm. (**o**) 11 August μm. (**p**) 12 August μm. (**q**) 11 September μm. (**r**) 12 September μm. (**s**) 11 October μm. (**t**) 12 October μm. (**u**) 11 November μm. (**v**) 12 November μm. (**w**) 11 December μm. (**x**) 12 December μm.

Table 3 shows the coefficients of the regression equation for each month. For 11 μm, the slope ranges from 0.9309 to 1.0009, and the intercept ranges from −0.0622 K to 18.8584 K. For 12 μm, the slope ranges from 0.9142 to 1.0061, and the intercept ranges from −1.6055 K to 23.0954 K. The intercept is greater in June and July. Using Equations (5) and (6), we obtained the thermal infrared channel $T_b$ data after MERSI calibration.

**Table 3.** The coefficients of the regression equation for each month.

| | 11 μm | | 12 μm | |
|---|---|---|---|---|
| | $K_1$ | $b_1$(K) | $K_2$ | $b_2$(K) |
| January | 0.9965 | 0.9709 | 1.0037 | −1.0263 |
| February | 0.9934 | 1.7541 | 1.0047 | −1.2531 |
| March | 0.9914 | 2.2408 | 1.0039 | −1.0523 |
| April | 0.9958 | 1.2124 | 1.0054 | −1.4132 |
| May | 0.9938 | 1.8305 | 0.9915 | 2.2519 |
| June | 0.9309 | 18.8584 | 0.9142 | 23.0954 |
| July | 0.9313 | 18.8522 | 0.9155 | 22.7913 |
| August | 0.9916 | 2.5781 | 0.9913 | 2.4517 |
| September | 1.0009 | −0.0622 | 0.9943 | 1.5972 |
| October | 0.9958 | 1.0709 | 0.9974 | 0.6583 |
| November | 0.9914 | 2.1522 | 1.0012 | −0.2765 |
| December | 0.9966 | 0.7916 | 1.0061 | −1.6055 |

### 3.3. Ice Surface Temperature Inversion Results

Sea ice forms in the Arctic when the temperature of seawater is below −1.8 °C. If there is no temperature constraint, the current temperature zone may also contain water. We assumed that the pixels with temperatures < −1.8 °C were ice and those with temperatures > −1.8 °C were open water. Not all clouds are detected by the cloud detection process, so some of the areas affected by clouds in the MODIS and MERSI data were not removed. The daily SIC product for the Arctic was retrieved from the FY-3B and FY-3D MWRI brightness temperature data. Using the method of calculating the SIC [28], spatiotemporal matching was performed between the SIC, MODIS and MERSI data. We used the area of the region with SIC > 15% and IST > −1.8 °C to calculate the IST for monthly error analysis. Additionally, we compared the calculated results with the MODIS MYD29 IST product and L4 IST data.

We selected the FY-3D/MERSI thermal infrared $T_b$ data in November 2020 to December 2021 to retrieve the IST in the Arctic based on the MODIS split-window algorithm of Equation (7). Figure 6 shows the strip result of the IST inversion result after spatial matching. Figure 7 compares the daily average MERSI IST inversion results with those of the MODIS MYD29 product.

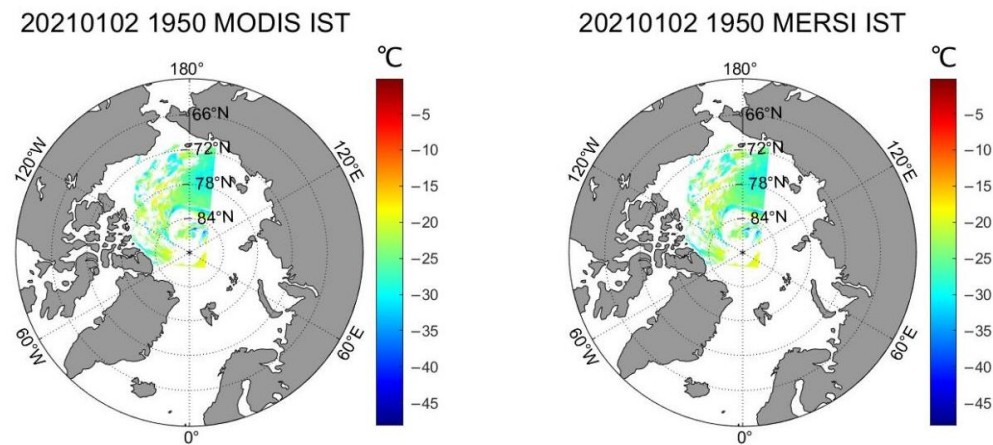

**Figure 6.** Comparison of the MODIS MYD29 product and the MERSI IST before calibration at 19:50 on 2 January 2021.

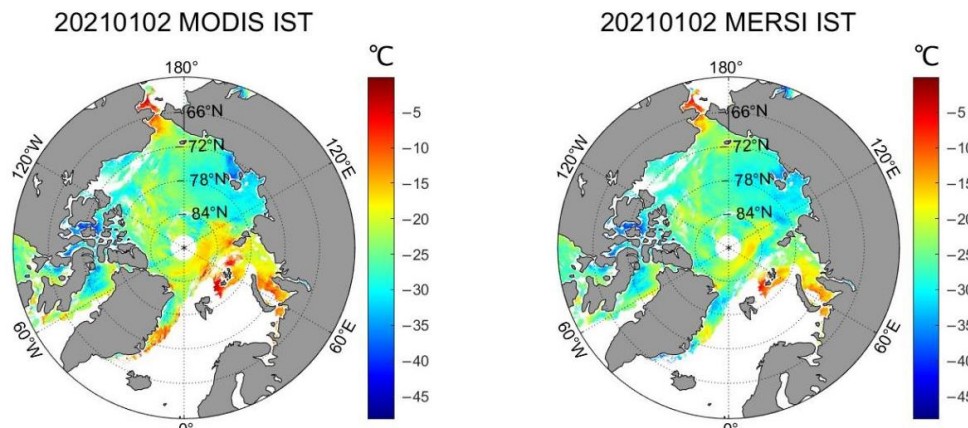

**Figure 7.** Comparison of distribution maps of the MODIS MYD29 IST product and MERSI inversion daily IST result before calibration on 2 January 2021.

## 4. Discussion

### 4.1. Analyzing Results of Brightness Temperature

In Section 3.2, we determined that the overall change in brightness temperature in the Arctic region shows a trend of increasing first and then decreasing. After fitting, the 11 μm and 12 μm channels of MERSI and MODIS data are in good agreement, and the scatter sets are distributed near the 1:1 line.

The cross-calibration parameters show that the slope of the calibration equation is about $1 \pm 0.05$ and the intercept is about $-1.6$–$2.6$ K, except in June and July (Table 3 and Figure 5). In June and July, the intercept is relatively large, the slope has a low value, and the overall brightness temperature is high, which may be due to the influence of summer clouds and water vapor.

In summer, melting decreases on the margin of the sea ice, there is a strong direct exchange of moisture between the sea and the atmosphere, and the amount of cloud increases; the cloud is the main factor affecting the identification of sea ice [32]. In this paper, the data after using the cloud mask are regarded as valid data, and the ratio between the valid data and total data is called the cloud-free rate. It can be seen from Table 4 that the cloud-free rate is related to the season. The cloud-free rate from January to April and from September to December is greater than 80%, while the cloud-free rate from May to August is relatively low. Because sea ice cover can vary from nearly 0% to 100%, it can provide different reflectance values and surface temperatures even within a single scene as a result of mixed-pixel effects [33]. Sea ice can also provide different reflectance values depending on the snow cover and the presence of melt ponds. The presence of melt ponds in the summer months will also affect the emissivity of the ice surface and therefore the calculation of the IST [33]. Figure 8 shows the comparison with daily average $T_b$ data for other months similar to Figure 4 (Section 3.1); there is only a small number of data and higher brightness temperature after the removal of clouds on 1 July 2021, which may affect the results of the cross-calibration. Most brightness temperatures in June and July are greater than 245 K, which is higher than those of other months (Figure 5k–n).

**Table 4.** The cloud-free rate for each month.

|  | Cloud-Free Rate |
| --- | --- |
| January | 94.85% |
| February | 93.98% |
| March | 94.00% |
| April | 86.69% |

**Table 4.** *Cont.*

|  | Cloud-Free Rate |
| --- | --- |
| May | 60.85% |
| June | 34.08% |
| July | 27.27% |
| August | 43.16% |
| September | 85.70% |
| October | 89.58% |
| November | 82.77% |
| December | 82.84% |

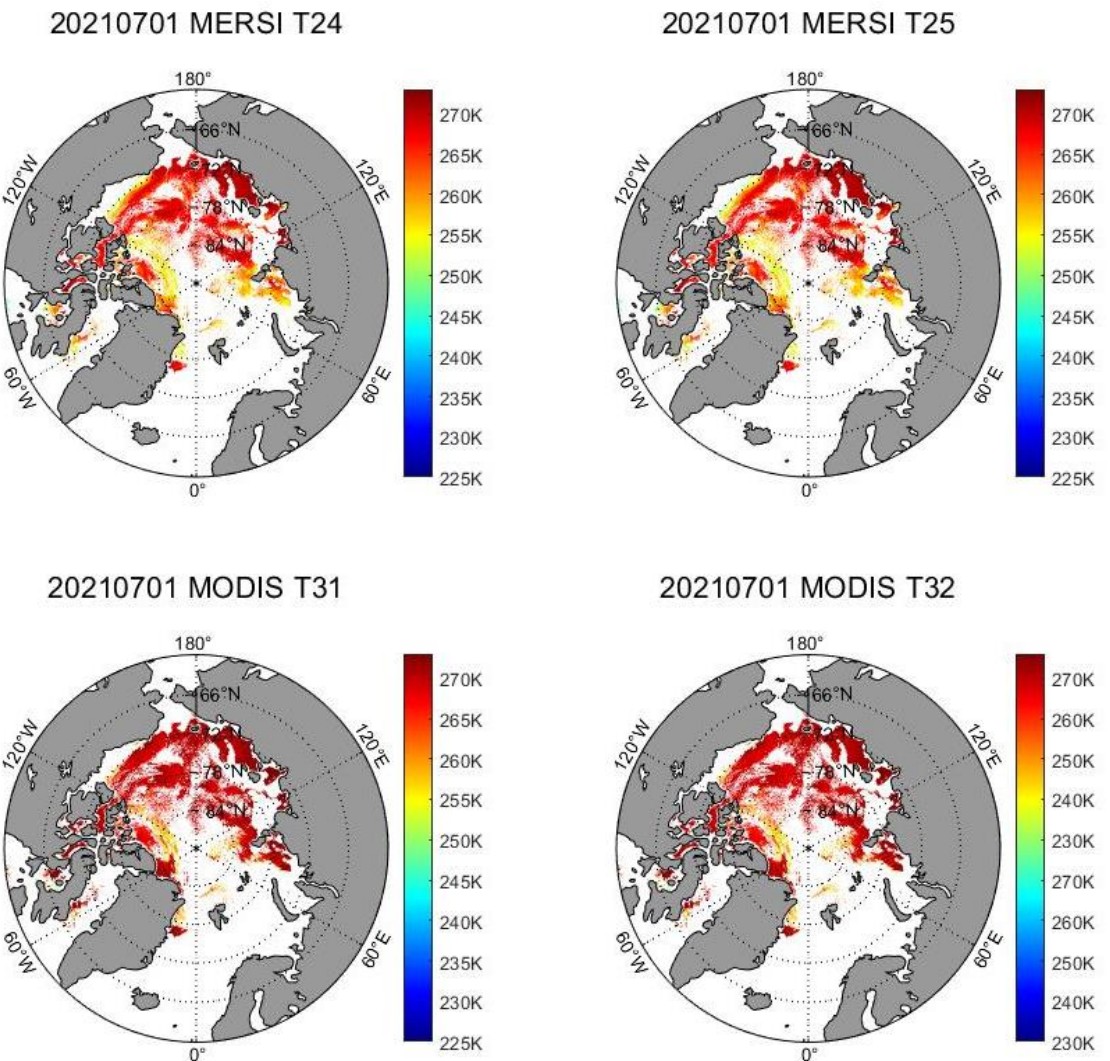

**Figure 8.** Comparison of distribution maps of MERSI (channels 24 and 25) and MODIS (channels 31 and 32) daily average thermal infrared $T_b$ data on 1 July 2021.

We compared the Std and bias of the brightness temperature between MERSI and MODIS in the 11 µm and 12 µm channels, respectively. It can also be seen from Figure 5 that the brightness temperature ranges of MERSI and MODIS are basically the same, and the mean bias between them is small. Table 5 shows that the distribution range of the thermal infrared 11 µm and 12 µm channels $T_b$ data from MERSI and MODIS is generally

consistent. The mean bias range at 11 μm was −0.5501–0.0560 K, and the Std was <1.2203 K. The mean bias range at 12 μm was −0.2591–0.1262 K, and the Std was <1.3582 K.

**Table 5.** Statistical values of $T_b$ data between channels 11 μm and 12 μm of MERSI and MODIS from January to December.

|  |  | **11 μm** | **12 μm** |
|---|---|---|---|
| Matching points | | 319,654−3,658,478 | |
| MERSI brightness temperature variation range (K) | | 219.4050–271.2189 | 218.8288–271.3484 |
| MODIS brightness temperature variation range (K) | | 217.4999–271.3484 | 217.1752–271.3449 |
| Mean bias (K) | Maximum | 0.0560 | 0.1262 |
|  | Minimum | −0.5501 | −0.2591 |
|  | Average | −0.1867 | −0.0253 |
| Standard deviation (K) | Maximum | 1.2203 | 1.3582 |
|  | Minimum | 0.7306 | 0.7967 |
|  | Average | 1.0320 | 1.1165 |

The brightness temperature results of each channel were further compared. Figure 9 shows the results of mean bias obtained from MERSI minus MODIS and Std of the 11 μm and 12 μm channels every month. The red line represents 11 μm, and the blue line represents 12 μm. As can be seen from Figure 9, the monthly data distribution of the 11 μm and 12 μm channels tends to be consistent, and the difference between the mean bias and Std of the two channels is small, with the difference between the mean bias of the two channels not exceeding 0.3 K, and the Std of bias not exceeding 0.2 K. From October to December in particular, the bias of 11 μm and 12 μm is very close.

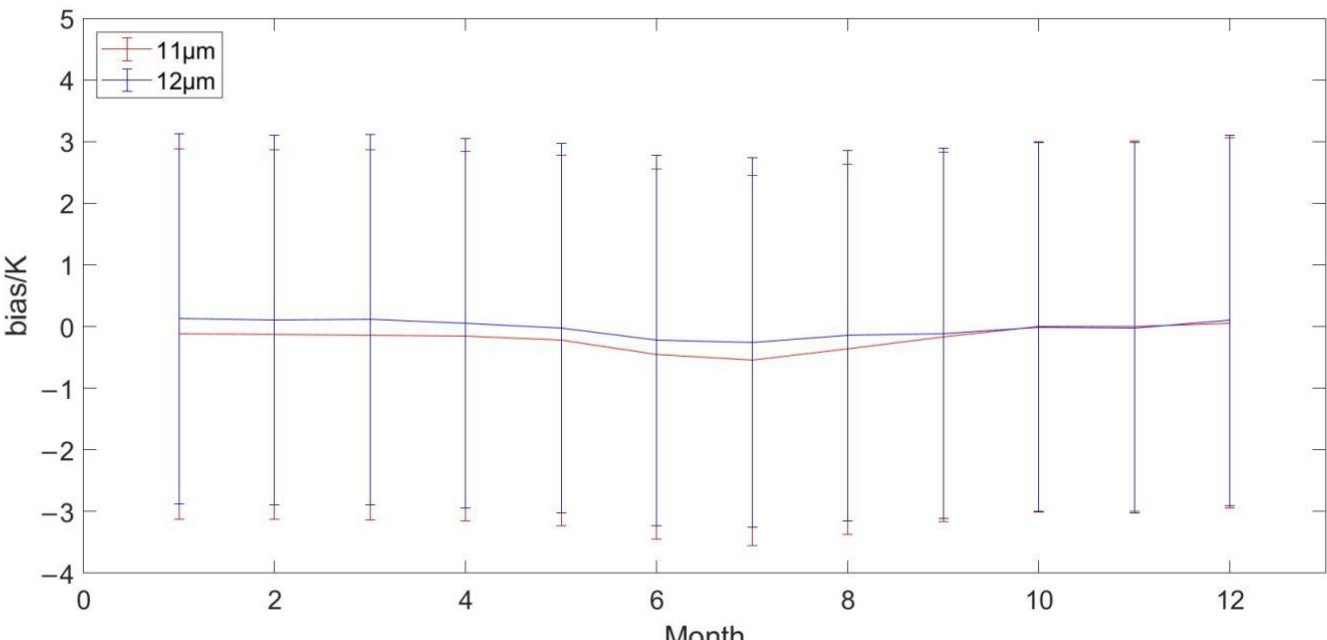

**Figure 9.** Mean bias and Std of the thermal infrared channels 11 μm and 12 μm $T_b$ data for MERSI and MODIS.

January to December monthly data are shown in Table 6, and MERSI has a high correlation with MODIS 11 μm and 12 μm channels, with correlation coefficients > 0.9553. After linear calibration, the bias and Std were reduced. The mean bias range of 11 μm

was $-0.0214 - 0.0105$ K, and the Std was <1.2190 K. The mean bias range of 12 μm was $-0.0096 - 0.0119$ K, and the Std was <1.2987 K. The mean bias in October, November, and December was small. The results show that calibration brings the MERSI channel brightness temperature closer to that of the MODIS channel and that the MERSI channel brightness temperature becomes more accurate, which is helpful for the IST inversion. We further analyzed the results of the whole year from January to December, and found that the annual mean bias of the 11 μm thermal infrared channel decreased from 0.1343 to $-0.0004$ K, and that of the 12 μm thermal infrared channel decreased from $-0.0488$ to $-0.0044$ K.

**Table 6.** Analysis of the monthly thermal infrared $T_b$ data between MERSI and MODIS before and after calibration.

| | | 11 μm | | 12 μm | |
|---|---|---|---|---|---|
| | **Matching Points** | **Standard Deviation (K)** | **Mean Bias (K)** | **Standard Deviation (K)** | **Mean Bias (K)** |
| | | Before/after | Before/after | Before/after | Before/after |
| January | 3658478 | 1.0238/1.0236 | −0.1207/−0.0079 | 1.0904/1.0902 | 0.1262/0.0073 |
| February | 2619298 | 1.0550/1.0540 | −0.1296/0.0038 | 1.1222/1.1217 | 0.1049/0.0064 |
| March | 3577109 | 1.0829/1.0816 | −0.1393/0.0090 | 1.1546/1.1543 | 0.1110/0.0082 |
| April | 2813160 | 0.8614/0.8608 | −0.1528/0.0037 | 0.9238/0.9230 | 0.0528/−0.0027 |
| May | 1536361 | 0.7306/0.7300 | −0.2234/−0.0132 | 0.7967/0.7953 | −0.0229/0.0085 |
| June | 487749 | 1.0290/0.9974 | −0.4496/−0.0138 | 1.1449/1.0968 | −0.2261/0.0101 |
| July | 319654 | 1.2034/1.1639 | −0.5501/−0.0214 | 1.3582/1.2987 | −0.2591/0.0041 |
| August | 373463 | 0.9020/0.9007 | −0.3683/−0.0086 | 1.0109/1.0091 | −0.1488/0.0062 |
| September | 763569 | 1.0110/1.0110 | −0.1691/0.0014 | 1.1108/1.1103 | −0.1121/0.0119 |
| October | 1308421 | 1.0843/1.0841 | 0.0015/0.0087 | 1.1600/1.1599 | −0.0098/−0.0096 |
| November | 1490283 | 1.2203/1.2190 | 0.0047/−0.0001 | 1.2793/1.2793 | −0.0198/0.0045 |
| December | 708691 | 1.1803/1.1800 | 0.0560/0.0105 | 1.2463/1.2454 | 0.0998/−0.0039 |

## 4.2. Analyzing the Results of the Ice Surface Temperature

### 4.2.1. Comparison with Data from MODIS MYD29 Product

The MERSI IST of thermal infrared channel $T_b$ data inversion was compared with the IST data of the MODIS MYD29 product (Table 7). The correlation between the MERSI and MODIS IST was >0.9572. The monthly mean bias shows that there was difference between the IST data retrieved from MERSI and MODIS MYD29 IST data. The monthly mean bias decreased from $-1.1303 - 0.0483$ °C to $-0.0612 - 0.0423$ °C and the Std was <1.3988 °C after calibration. In Table 7, it can be seen that the IST mean bias in October, November, and December was still small. Figure 7 (Section 3.3) and Figure 10 compare the daily average MERSI IST inversion results before and after calibration with that of the MODIS MYD29 product on 2 January 2021.

**Table 7.** Analysis of the monthly FY-3D/MERSI and Aqua/MODIS IST data before and after calibration.

|  | Standard Deviation (°C) | Mean Bias (°C) | Corr |
|---|---|---|---|
|  | Before/after | Before/after |  |
| January | 1.2399/1.2392 | −0.5166/−0.0283 | 0.9777 |
| February | 1.2537/1.2429 | −0.5042/0.0024 | 0.9844 |
| March | 1.2669/1.2563 | −0.5378/0.0156 | 0.9794 |
| April | 1.0639/1.0565 | −0.4873/0.0134 | 0.9904 |
| May | 0.9481/0.9460 | −0.5952/−0.0479 | 0.9852 |
| June | 1.1254/1.0869 | −0.8963/−0.0530 | 0.9572 |
| July | 1.2116/1.1758 | −1.1303/−0.0612 | 0.9656 |
| August | 1.0358/1.0326 | −0.7909/−0.0307 | 0.9878 |
| September | 1.1532/1.1521 | −0.2730/−0.0124 | 0.9781 |
| October | 1.2618/1.2588 | 0.0242/0.0423 | 0.9736 |
| November | 1.4118/1.3988 | 0.0483/−0.0085 | 0.9788 |
| December | 1.4027/1.3942 | −0.0129/0.0367 | 0.9843 |

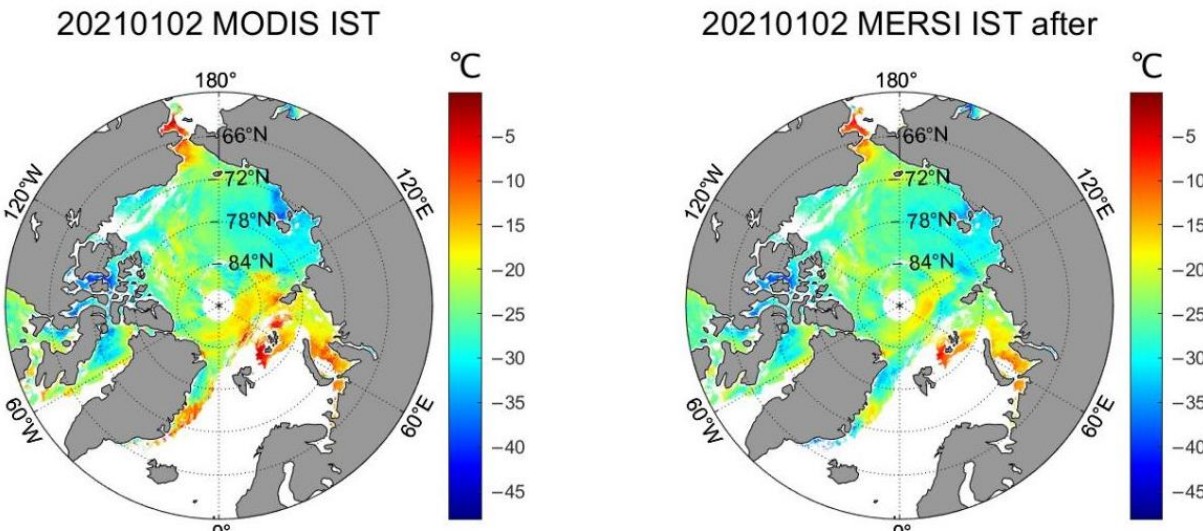

**Figure 10.** Comparison of distribution maps of the MODIS MYD29 IST product and MERSI inversion daily IST result after calibration on 2 January 2021.

We further compared the annual IST data of MERSI after calibration and MODIS in Figure 11; the scatter sets are distributed near the 1:1 line. Through data analysis, the annual mean bias of the MODIS and MERSI IST decreased from 0.4367 to 0.0036 °C. The MODIS IST is slightly larger than the MERSI IST. The IST of MERSI after calibration is closer to that of MODIS, which ensures that most of the IST accuracy ranges are consistent with MODIS. In Figure 12, we analyzed the statistical histogram distribution of the MODIS and MERSI IST before and after calibration from January to December. The red line represents MERSI, and the blue line represents MODIS. This calibration makes the MERSI IST more accurate and closer to the MODIS MYD29 IST product data.

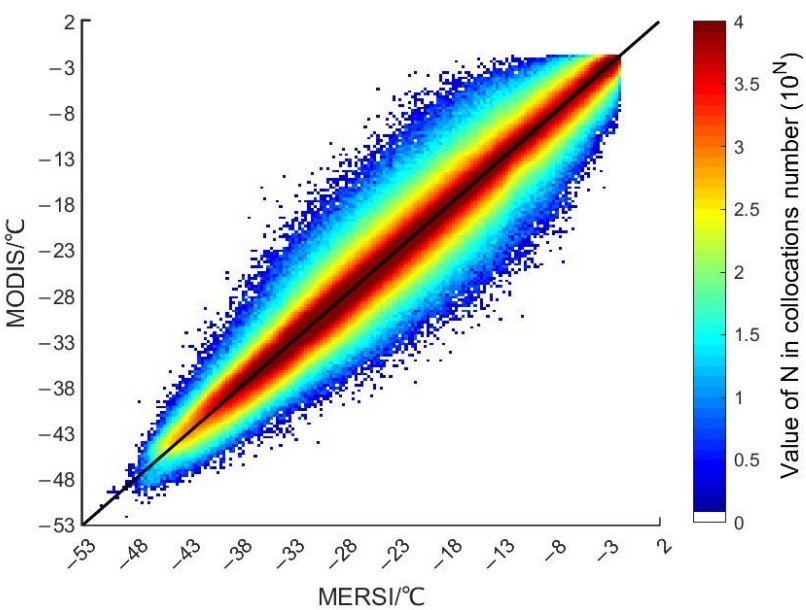

**Figure 11.** Scatter plot diagram of MERSI and MODIS IST after calibration from January to December.

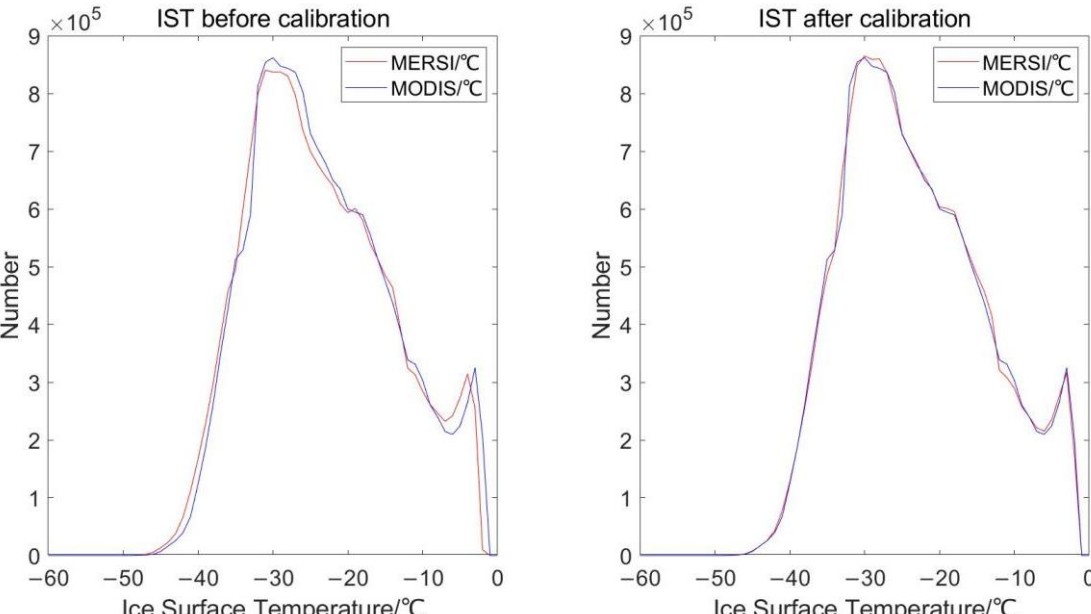

**Figure 12.** Statistical histogram distribution of the MODIS and MERSI IST before and after calibration from January to December.

### 4.2.2. Comparison with Data from L4 IST

L4 IST data from January to May 2021 were used as the comparison data. Because L4 IST data are only available until May 2021, the MERSI IST data were also from January to May 2021. Then, we performed a statistical analysis based on the spatiotemporal matching results of the MERSI IST and L4 IST data. In general, there is a consistency between the MERSI IST and L4 IST data (Figure 13).

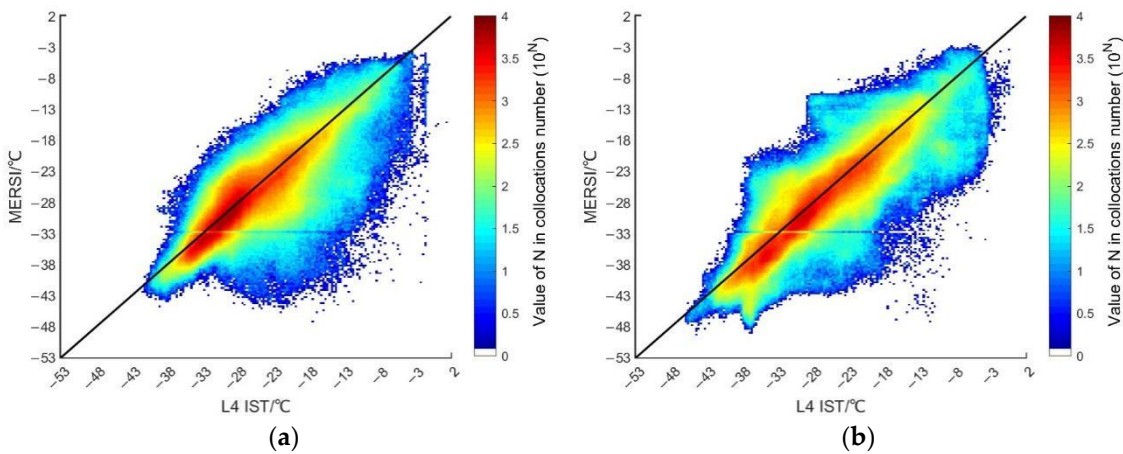

**Figure 13.** (**a**) Scatter plot diagram of L4 IST and MERSI IST data in January. (**b**) Scatter plot diagram of L4 IST and MERSI IST data in February.

The MERSI IST of thermal infrared channel $T_b$ data inversion was compared with the L4 IST data, as shown in Table 8. The correlation between the MERSI IST and L4 IST was >0.7844. From January to May, the monthly mean bias of MERSI IST and L4 IST data was $0.9891 - 2.7510\ ^\circ\text{C}$, and the Std was <3.5774 $^\circ$C. The overall bias of MERSI IST and L4 IST data for five months was 1.8006 $^\circ$C, the Std was 3.378 $^\circ$C, and the correlation was 0.9235.

**Table 8.** Analysis of the monthly L4 IST and FY-3D/MERSI IST data.

|          | Standard Deviation ($^\circ$C) | Mean Bias ($^\circ$C) | Corr   |
| -------- | ------------------------------ | --------------------- | ------ |
| January  | 3.2689                         | 0.9891                | 0.8341 |
| February | 3.3007                         | 1.0601                | 0.8941 |
| March    | 3.0176                         | 2.2137                | 0.8679 |
| April    | 3.5774                         | 2.7510                | 0.8784 |
| May      | 3.3900                         | 2.6514                | 0.7844 |
| Total    | 3.3780                         | 1.8006                | 0.9235 |

## 5. Conclusions

Through the comparative analysis of channels 24 and 25 of FY-3D/MERSI and channels 31 and 32 of the Aqua/MODIS over 12 months, it can be seen that there was a strong correlation between the Aqua/MODIS and FY-3D/MERSI thermal infrared $T_b$ data, with correlation coefficients >0.95. The bias between MERSI and MODIS $T_b$ data was $-0.5501$–0.1262 K. Compared with other months from January to December, the mean biases in October, November and December were smaller, less than 0.0998 K. We cross-calibrated the data for the monthly Arctic thermal infrared $T_b$ data obtained by channels 24 and 25 of the FY-3D/MERSI and channels 31 and 32 of the Aqua/MODIS. After calibration, the bias was $-0.0214$–0.0119 K.

We used the split-window algorithm to calculate the IST obtained when using MERSI thermal infrared $T_b$ data inversion before and after calibration, and we conducted a comparative analysis of the IST data and MODIS MYD29 IST product. The analysis shows that it is feasible to cross-calibrate the Aqua/MODIS and FY-3D/MERSI data. The mean bias and the Std of the Aqua/MODIS and FY-3D/MERSI are lower after calibration than before calibration. The monthly mean bias decreased from $-1.1303$–0.0483 $^\circ$C to $-0.0612 - 0.0423\ ^\circ\text{C}$ and the Std was <1.3988 $^\circ$C after calibration. The errors after calibration are also reduced, and the MERSI IST inversion results are more accurate than before calibration. The research results of this paper prove that the quality of FY-3D/MERSI data is very good and that the mean bias of data in individual months is maintained in a small range after cross calibration. FY-3D/MERSI infrared data can be used to provide parametric support for Arctic applications.

The MODIS MYD29 product is available from NASA's Atmospheric Archives and Distribution System Web site and has been verified. The comparative analysis of MODIS and MERSI IST shows that the quality of MERSI IST is comparable to that of MODIS. At the same time, the IST obtained when using the calibrated MERSI thermal infrared $T_b$ data inversion and the L4 IST data released by Copernicus Marine Service were verified. The results showed that the L4 IST and MERSI IST data were consistent. From January to May 2021, the monthly mean bias was 0.9891–2.7510 °C, and the Std was <3.5774 °C. There were no IST data in many areas of the Arctic due to the influence of clouds, and the MODIS cloud mask was used as the MERSI cloud detection result, which means there is a lack of data in the MERSI IST inversion results. Microwave data are not affected by clouds and can be used as supplementary data, so the fusion of IST data retrieved using thermal infrared and microwave channels data is our next direction of research.

**Author Contributions:** Conceptualization, H.C., X.M., L.L. and K.N.; methodology, H.C. and X.M.; formal analysis, X.M. and H.C.; investigation, H.C., X.M., L.L. and K.N.; data curation, H.C., X.M., L.L. and K.N.; writing—original draft preparation, X.M. and H.C.; writing—review and editing, H.C., X.M. and L.L.; funding acquisition, H.C. and L.L. All authors have read and agreed to the published version of the manuscript.

**Funding:** This research was funded by the National Key R&D Program of China, grant number 2019YFA0607001.

**Institutional Review Board Statement:** Not applicable.

**Informed Consent Statement:** Not applicable.

**Data Availability Statement:** FY-3D/MERSI brightness temperature data are provided by China National Satellite Meteorological Center (http://www.nsmc.org.cn/ (accessed on 12 January 2022)). The MODIS data are available from NASA's Atmospheric Archives and Distribution System Web site (LAADS Web; http://ladsweb.nascom.nasa.gov/ (accessed on 27 January 2022)). The Sea Ice Concentration data released in the process of the polar sea and key laboratory of changes in the global Marine website (http://coas.ouc.edu.cn/pogoc/sy/list.htm (accessed on 20 Octorber 2022)). L4 SST data are provided by Copernicus Marine Service (https://data.marine.copernicus.eu/product/SEAICE_ARC_PHY_CLIMATE_L4_MY_011_016/description (accessed on 16 November 2022)).

**Acknowledgments:** This work was supported by the National Key Research and Development Program of China 2019YFA0607001. The MERSI data were provided by the NSMC Satellite Data Center.

**Conflicts of Interest:** The authors declare no conflict of interest.

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
