# Peer review of "Quality Assessment of FY-3D/MERSI-II Thermal Infrared Brightness Temperature Data from the Arctic Region: Application to Ice Surface Temperature Inversion"

_remotesensing, doi:10.3390/rs14246392_

Round 1
Reviewer 1 Report
The study assessed the observations of temperature data by FY-3D/MERSI-II in the Arctic region and applied the obtained algorithm to identify the surface temperature of sea ice. The topic is very important and the study provide great insight on obtaining the satellite observed dataset in the Arctic Ocean. I am generally satisfied with the quality of the manuscript and some minor modifications are required for further clarification and improvement.
Firstly, all the measurements are from satellite, which can introduce a system error for the observations. Some in-situ measurements from buoy and ITP are more helpful for improving the assessment of the datasets.
Also, the MODIS onboard Aqua is applied in this study, but data quality for the temperature over ice can be difficult to be validate. The MODIS onboard Terra can also be applied for cross validation of the algorithm.
Secondly, obtain dataset over a long period, i.e., at least for one year, and calculating the number with available observations and ratio of cloud free rate. Ideally, the coverage rates in different seasons are more important, which is necessary for readers to assess the possibility to use satellite dataset.
Thirdly, convert the unit from Calvin to Celsius that is easier for understanding the environmental conditions in the Arctic Ocean.
Lastly, the citations are not consistent. For example, only the first letter of the last name should be capitalized.
Author Response
请参阅附件。

Reviewer 2 Report
Comments to the Author.
This paper explored the temporal and spatial variations in the brightness temperatures with FY-3D/MERSI-II and MODIS profiles. the authors have also investigated the intercomparisons between various regions in the Arctic region north of 60 ° N, as shown in Figures 4 and 5. The importance of this study to the satellite remote sensing community is the very detailed approach of using model simulated TOA radiances and careful selection of matchup observations of other satellites that is required for validation of satellite derived end products such as IST. However, the method was not well-described, and the result analyses were not very clear. Also, almost every paragraph has some grammar errors. Also the authors didn’t add the line number to the text. It’s hard to give comments based on page number. Thus, the manuscript should major-review before it is published.
Major Comments.
1. The equations of and 6 are the primary method in the manuscript. However, it is quite a lot unclear. First, we do not know where the equation came from. The reference [6], which the author provided, we cannot find the paper at Google scholar. The author should provide a detailed description and reference about how to get this equation. Second, how to define the coefficients K and b, how to do the regression, and what data are used for the regression should be discussed as they are the proposed method in the manuscript.
2. It was not clear to me whether potential sunglint affected pixels were identified and rejected from the matchups. Also, I did not know if any of the cloud and ice water and cloud diagnostic product information were considered for potential usefulness related to identifying scenes potentially affected by thin cirrus.
Minor Comments.
English editing required
There are lots of errors related with the typo, overall expression in this article seems not to be scientific and professional. The author should define abbreviations and acronyms the first time they are used in the text. Because I can’t get the line number so please use the search function to find that.
“on the Arctic ice was evaluated” should be “on the Arctic ice were evaluated”
“ after calibration is better than” should be “ after calibration are better than”
which obtained by the split window “ ” should be “which was obtained by the split window ”
“ also has reference value” should be “ also has reference values”
“is split from ” should be “ are split from”
“ MYD021 datasets includes the” should be “MYD021 datasets include the ”
“ period of MERSI and MODIS strip data are 5 minutes” should be “period of MERSI and MODIS strip data is 5 minutes ”
“The Bias in October, November and December are small. ” should be “The Bias in October, November and December is small. ”
Abbreviation microwave radiance imager (MWRI) was defined in page 2 and no need to define again in page 4.
